# What’s in a Name? An Overview of the Proliferating Nomenclature in the Field of Phage Lysins

**DOI:** 10.3390/cells12152016

**Published:** 2023-08-07

**Authors:** Roberto Vázquez, Yves Briers

**Affiliations:** Laboratory of Applied Biotechnology, Department of Biotechnology, Ghent University, 9000 Ghent, Belgium

**Keywords:** bacteriophage, lysins, phage lysins, enzymes, antimicrobials, nomenclature

## Abstract

In the last few years, the volume of research produced on phage lysins has grown spectacularly due to the interest in using them as alternative antimicrobials. As a result, a plethora of naming customs has sprouted among the different research groups devoted to them. While the naming diversity accounts for the vitality of the topic, on too many occasions it also creates some confusion and lack of comparability between different works. This article aims at clarifying the ambiguities found among names referring to phage lysins. We do so by tackling the naming customs historically, framing their original adoption, and employing a semantic classification to facilitate their discussion. We propose a periodization of phage lysin research that begins at the discovery era, in the early 20th century, enriches with a strong molecular biology period, and grows into a current time of markedly applied research. During these different periods, names referring to the general concepts surrounding lysins have been created and adopted, as well as other more specific terms related to their structure and function or, finally, names that have been coined for the antimicrobial application and engineering of phage lysins. Thus, this article means to serve as an invitation to the global lysin community to take action and discuss a widely supported, standardized nomenclature.

## 1. Introduction

Research on “phage lysins” is a hot topic these days: in this so-called “post-antibiotic” world to come, the questions and answers on how to handle antibiotic-resistant bacteria are one of the most apparent drivers for innovation in antimicrobial discovery. Under the influence of this contemporary challenge, bacteriophage research has drifted towards an eminently applied focus of breadth and intensity unheard of (in the West) since the early days of phage. That is how we end up today with a globally growing community of researchers putting their efforts not only on the development of full virion phage therapy [1,2] but also on repurposing phage enzymatic products as antibacterial drugs [3,4,5]. This latter approach has quickly gained relevance since some seminal papers published in the early 2000s by the V. Fischetti group at Rockefeller University [6,7,8]—although the general idea of using enzymes as therapeutic agents was already present in much earlier literature [9,10]. The interest in these enzymatic phage products —which are usually referred to as “lysins”—has spectacularly grown in the last few years, probably due to two factors: (1) the purely scientific interest sparked by the versatility and amenability of this kind of proteins for engineering, biochemical, functional or evolutionary studies [11,12,13,14], as well as by the massive amount of (sequence) data and prior knowledge publicly available [15,16,17] and (2) the feasibility of lysins as a new kind of antibacterial that can fit within the regulatory, industrial, and clinical frameworks and pipelines already established for the production of therapeutic proteins (biologicals) [18]. In this perspective, it is relevant to note that, according to the current literature, it can be said that phage lysins fulfill the four innovation criteria defined by the World Health Organization: being a new class of antibiotics, having a new mode of action, a new target, and lack of known cross-resistance [19].

All of these new research efforts come with their own nomenclature innovations as much as they have continued the naming traditions of the field, with diverse emphases and novel uses. Therefore, we stand at a point where names in the phage lysin field proliferate and are sometimes used too flexibly, which, in turn, creates a sometimes -ambiguous research ecosystem, which is difficult to penetrate for the lysin-layman and exhausting to navigate for experts. Indeed, the naming proliferation reflects the vitality of the phage lysin field. However, it also faces us with what can become an uncontrollable Tower of Babel, in which retrieving and comparing data from all the international stakeholder groups becomes increasingly complicated. This article thus intends to establish the basis for an ordered discussion on the naming customs in the phage lysin field.

If we are to discuss “customs”, the appropriate way to do so is by relying on the historical circumstances that explain them. At least three “eras” can be clearly described in the history of lysin nomenclature, and each of them encompasses a set of names and concepts devised within it (Figure 1).

(1)Discovery era: roughly between the early 20th century and the 1940s, the fundamental terminology in the field was developed at the same time that phenomena were first time described.(2)Molecular era: in the central and final years of the 20th century, phage research entered the molecular biology era in which the infection and enzyme activity mechanisms were elucidated.(3)Applied era: since the beginning of the 21st century, phage lysin research entered a more markedly applied stage driven by the antibiotic resistance crisis. At this point, more distinctions came into place, many of which are related to the new role of lysins as antimicrobials. Even more remarkable is the plethora of common names given to the vast number of new lysins being described and investigated, both wild type and engineered, following the most diverse reasonings.

In this work, lysin-related names will be thematically reviewed, attending to their different semantic levels: (1) general terminology for lysins, (2) terms related to the structure and function of lysins, (3) names for lysins as antibacterial agents and (4) nomenclature of individual lysins. For each case, the appropriate historical background will also be provided, considering the periodization established above (Figure 1).

## 2. General Terminology for Lysins

According to our PubMed-based survey (Table 1), the terms “lysin” or “endolysin” are the most popular names given to these enzymes encoded in phage genomes whose main function is to actively and directly cause bacterial host cell lysis at the end of the replication cycle. “Lysin” is indeed a rather old name, which can be traced back, in the phage context, to the days of Felix d’Herelle [20] (see Appendix A for an English translation of the earliest example of the use of the term “lysin” applied to a suspected enzyme accompanying the “bacteriophage phenomenon”) and Frederick Twort’s [21] pioneering research. However, given the still unclear identity and molecular basis for the “bacteriophage phenomenon” at the time, the term “lysin” was still confusedly applied in general to observations of “clearance” in the field of immunology, bacteriology or the incipient virology. This eminently occurs in the early articles by Twort (see e.g., [21]), in which he refers to the bacteriophage (still hesitantly identified by him as an “ultramicroscopic virus”) rather indistinctly as “lysin”. Even in a later stage, when the identity of the phage as a replicating virus was more or less clear and the original distinction posed by d’Herelle between a non-transmissible lysin and a transmissible virus was somewhat adopted, “lysin” was still confusedly used for observations linked both with the peptidoglycan degradation activity—the proper cell lysis—and the action of phage depolymerases [22]. When it was made clear that the action of depolymerases was separate from the key role of lysins in the phage infective cycle, the name “lysins” mostly acquired the meaning of “peptidoglycan-degrading enzyme”. From then on, any other non-enzymatic player in host lysis (holins, pinholins, spanins, or releasins) was left out of the “lysin” class (although these proteins were only discovered later [23,24,25]). Thus, lysins are also referred to as peptidoglycan-degrading enzymes or their pure synonym, “muralytic enzymes”. Taking these distinctions into account, “phage lytic enzyme” can function as a synonym of lysin, but “phage lytic protein” is ambiguous since the non-enzymatic agents of phage-mediated lysis can easily be included under this name.

Of note, phage depolymerases (more accurately “exopolysaccharide—EPS—depolymerases” and also anecdotally known as “biofilm-disrupting enzymes”) have also been considered as “phage lytic enzymes” [26], although this may also be a remnant of the times when some confusion still existed between the actual lawn clearance of plaques and depolymerase-induced halos.

The similar but more meaningful term “endolysin” seems now to be preferred to the simpler “lysin” (Table 1). This name dates back to the mid-20th century [27,28,29,30] as applied to phage lysins (it was earlier—although fleetingly—used to designate a bacteriolytic substance “present in extracts of leukocytes” [19]). Of no surprise, this is also the time in which the specific role of the endolysin as an unequivocally enzymatic player, key to the release of the phage progeny—or lysis “from within”—was elucidated. Despite the somewhat “liberal” current use to designate lysis-exerting, peptidoglycan-degrading, phage-encoded enzymes, the strict etymological sense of “endolysin” raises a problem. Given its Greek meaning (“lysis from within”), it should be preferentially reserved to the lysis-inducing enzyme that is produced within the host and released towards the cell wall at the end of the phage lytic cycle. Nonetheless, it is now (misleadingly?) used to refer both to the native endolysins found at the lytic cassette of phage genomes as much as to synthetic or engineered lysins that cannot be longer identified as an enzyme with such a physiological function. On the other hand, the preference towards “endolysin” may be explained because the name “lysin” can be equivocal, especially in the context of automatic database searches or in communicating science to the laymen (i.e., it is too similar to the amino acid “lysine”). The phrase “phage lysin”, however, preserves the meaning (lytic enzyme from phage origin) while avoiding misunderstandings and the misnaming of any peptidoglycan-degrading enzyme from phage origin as an endolysin.

This is important because phages do not only encode lysins at their lytic cassette, i.e., not all phage lysins are, in fact, endolysins. Since the mid-20th century, a muralytic activity was known to be present within the viral particles of some phages [31]. At that time, such an enzyme was only given a generic name, such as “lytic enzyme”, “lysin”, or even more often, “lysozyme” (as it will be later discussed). Specific names to distinguish true endolysins from these “structural lysins” probably only came into place when their widespread presence was experimentally established in the 1990s–2000s [32]. There is now an astonishing diversity of names for “structural lysins”. This latter term, for example, indicates that structural lysins are embedded within proteins that are a part of the virion structure and seem like a rather simple naming variation with little confusion risk at the moment. Other names emphasizing the structural character of these lysins are: “virion-associated peptidoglycan hydrolases” (VAPGHs or VAPHs, a disputable name on the basis of inaccurate description of the possible enzyme catalytic mechanism), “virion-associated lysins” (VALs), “tail-associated muralytic enzymes” (TAMEs) or “tail-associated lysins” (TALs). Another imaginative variant is “ectolysins”, which is indeed adequately derived as opposed to “endolysin”, emphasizing, in this case, that this kind of enzymes exerts lysis of the bacterial cell from without, although it is not widely adopted.

## 3. Terms Related to the Structure and Function of Lysins

While, in the beginning, the actual point of discussion was the nature of the phage itself as a differentiated biological entity, this became clearer with the development of electron microscopy and its early application to the “bacteriophage phenomenon”, which allowed identifying new “sperm-shaped” particles (as worded by Helmut Ruska in 1941 [33]) that, in turn, would be identified as bacterial viruses [34]. Thus, after the first wave of phage research, the field naturally evolved towards what is now known as molecular biology. In fact, many of the discoveries surrounding phages—and even many of the researchers involved in them—became the very basis for the conceptual construction of the new molecular discipline. This is how such features as molecular identity, genomic location and—even more relevant for our topic—catalytic specificity, started to receive the spotlight in new developments.

In this respect, one of the earliest names given to phage lysins was “lysozyme”. Controversy over correct nomenclature has spurred over time with this name. It was originally coined by Alexander Fleming in a seminal paper in which he thoroughly described the bacteriolytic effects of human secretions—thus first describing this bacteriolytic enzyme as a component of innate immunity [10]. Later, the term was established to refer to any bacteriolytic enzyme whose catalyzed reaction resulted in the release of glucosamine and muramic acid and the liberation of reducing groups [35]. Koch and Dreyer (1958) [36], elaborating on this somewhat liberal definition, labelled a phage lysin (that of T2 coliphage) for the first time as a lysozyme (to our knowledge). Although the authors were indeed cautious with their wording (“the data presented show that the action of the phage enzyme is analogous to that of egg white lysozyme. It is suggested, therefore, that this enzyme be termed “phage lysozyme”. Enzymes found in other phage systems may have similar specificity”) we believe this was the beginning of the generic use of “lysozyme”. This terminology rapidly spread to the rest of the lysins from the most intensely-studied phages at that time: the coliphages (T-series phages or the lambda phage). With the systematization of enzyme nomenclature, “lysozyme”, and its then synonym “muramidase”, ended up being just the names given to a specific type of enzymes catalyzing the hydrolysis of β-1,4-linkages between *N*-acetylmuramic acid and *N*-acetyl-d-glucosamine residues in peptidoglycan (as the original lysozymes from innate immunity). However, “lysozyme” was still being metonymically and traditionally applied to peptidoglycan-degrading enzymes with other catalytic specificities and mechanisms. The historical tradition of “lysozyme” can clearly be appreciated in the numbers found in Table 1, which make “lysozyme” and “muramidase” the most historically used terms referring to enzymatic activity in phage research. However, Figure 2 also shows that those terms were intensively used in the mid-20th century but are currently becoming less used in favor of “lysin” and “endolysin”. In fact, when the terms “T4”, “T7” and “lambda” are explicitly removed from the PubMed query, the 1202 and 1215 references found mentioning, respectively, “lysozyme” and “muramidase” in a phage context, become 396 and 443. This supports that “lysozyme” and “muramidase” were indeed used in the mid-20th century as some sort of inertia from the naming of the first lysins from the T-series and lambda phages, but were progressively—albeit slowly—removed when accurate description of catalytic specificities started to become more popular.

The establishment of “lysozyme” as just the name for muramidase activity created many nomenclature dissonances in the phage field, as has already been pointed out [37]. One of the most blatant examples is that of the so-called T7 lysozyme [38], which, in fact, is an *N*-acetylmuramoyl amidase (a name sometimes abbreviated as “NAM-amidase” or just “amidase”) that breaks the peptidoglycan between an *N*-acetylmuramoyl residue acid and the first l-amino acid residue of the stem peptide. It is thus not a true lysozyme. The name T7 lysozyme is still routinely used, for example, in the context of the T7 inducible expression systems, even if Inouye et al. (1973) already proved its different catalytic specificity and implicitly called the name into question [39]. The case of lambda phage lysozyme is also worth mentioning, since the traditional and still used “lysozyme” name is not completely accurate, given that it has been shown to preferably act as a lytic transglycosylase [40,41]. Of note, both T4 lysozyme (a true, canonical lysozyme) and lambda phage lytic transglycosylase are now clustered together in the same Pfam family, which is problematically named *Phage_lysozyme* (PF00959). All these lysozyme-like enzymes share homologous structural features and are indeed thought to be part of the same evolutionary family. In fact, the line between canonical, hydrolytic lysozymes and lytic transglycosylases seems rather blurred. For example, the hen egg white lysozyme has been shown to display both hydrolytic activity and transglycosylation [42,43], and while wild-type T4 lysozyme is not capable of transglycosylation, one or a few point mutations are enough to convert its activity towards lytic transglycosylation [44], which also explains the co-existence of preferred lysozymes, and preferred transglycosylases within its family. As a conclusion, a nomenclature based on the catalytic mechanism should be strictly based on direct experimental evidence, following the current recommendations by the International Union of Biochemistry and Molecular Biology (IUBMB), and in the absence of this kind of evidence a more generic name (e.g., lambda phage endolysin, T7 endolysin, etc.) should be preferred. Then, although nowadays the use of “lysozyme” as an umbrella term is scarce, this usage should indeed be avoided so as not to create any more confusion, therefore reserving the term (and its synonym, “muramidase”) for those lysins whose catalytic specificity is the one intended by its currently accepted definition (EC 3.2.1.17, “hydrolysis of (1→4)-β-linkages between *N*-acetylmuramic acid and *N*-acetyl-d-glucosamine residues in a peptidoglycan and between *N*-acetyl-d-glucosamine residues in chitodextrins”). The same should apply to other names regarding the catalytic specificity (endopeptidase, glucosaminidase), for which an accepted definition is given in Appendix B.

In yet another turn of the screw, there is one more general term presumedly equivalent to “lysins” or “peptidoglycan-degrading enzymes”: the name “peptidoglycan hydrolases”, often abbreviated as PGHs, or the synonym “murein hydrolases”—since murein is another name for peptidoglycan. However, this may also be conflicting. Hydrolases are enzymes that use water to catalyze the breakage of a chemical bond. While most of the known catalytic diversity of lysins are, at least as predicted, hydrolases [17] it is quite apparent as already shown that many do not use such catalytic mechanisms for breaking peptidoglycan, but instead are “lytic transglycosylases”. Lytic transglycosylases achieve the breaking of glycan bonds by a mechanism other than hydrolysis (they can be thus considered “lysases”), in which they transform one glycoside into another without the intervention of a water molecule. Therefore, not all endolysins are conceptually comprised under the term PGHs, although some taxonomical turmoil persists given the fact that “transglycosylases” are classified as glycoside hydrolases into various GH (= “glycoside hydrolase”) families on the basis of structural similarity (see previous paragraph and CAZYpedia, https://www.cazypedia.org/index.php/Transglycosylases (accessed on 5 August 2023).

The issue can be further complicated in the case of mycobacteriophages. These phages comprise a couple of so-called “lysins” with different catalytic specificities. While mycobacteriophage lysin A can fall under the peptidoglycan-degrading paradigm presented for all “lysins”, lysin B, in fact, has the function of detaching the arabino-mycolyl outer layer of mycobacteria (or other Corynebacteriales species). As such, lysins B are, in fact, esterases that cleave the arabino-mycolyl ester bond. While it may be correct to denominate lysins B as “(endo)lysins” since they are indeed directly and enzymatically involved in bacterial cell lysis, it is less clear that they would fit under the terms “peptidoglycan-degrading enzyme” or PGHs. As esterases, they perform a hydrolytic reaction, thus they are hydrolases, but it is rather inaccurate to say that they degrade the peptidoglycan.

With the spectacular development of lysin combinatorial engineering from the beginning of the 21st century, the terms referring to the “building blocks” of lysins have also become increasingly relevant. The “mosaic” nature of phage genomes has been known since the second half of the 20th century. In fact, the realization that phage genomes are made up of DNA stretches that are conserved across many different phages [45], even traceable to bacterial genomes, got phage researchers to propose a “modular theory of phage evolution” [46]. According to it, phage genomes are the result of the joint evolution of sets of functionally and genetically interchangeable elements, which are known as “modules”. The modules that build up phage genomes can be full multi-gene cassettes (e.g., lytic cassettes), single genes or even only parts of the genes. The latter is the case for phage lysins, whose modular nature was early evidenced, for example, in a series of seminal works on the endolysins from *Streptococcus pneumoniae* phages [47,48]. The possibility of building chimeric lysins combining modules was also realized at that time [49], and since then it has become the main paradigm for lysin engineering [11]. Within this context, the terms “module” and “domain” are rather indistinctly used to refer to those functional building blocks that are exchanged and fused to make up new, even purely synthetic lysins. In the particular case of phage lysins, it may be acceptable to use “module” and “domain” as synonyms, but the nuances of each term are different:We may say that the main features of a module are two: (1) its ability to carry out its function in a relatively autonomous way, and (2) its interchangeability or possibility to be compatibly integrated into other higher-order units comprising different combinations of modules [46];The domain, on the other hand, simply means a structural, functional or evolutionary unit of a protein [50].

On this basis, and although there is much debate in the literature on the proper definitions, we may say that the main difference between these concepts lies in the “interchangeability” or “mobility” that comes with the term “module”. In any case, we do not see any conflict in using both terms interchangeably in the field of phage lysins.

Lysins are therefore composed of different protein domains, which behave as modules that can be exchanged naturally, as a part of the evolutionary process at the bacteria/phage interplay, or artificially to engineer new lysin variants. Some lysins contain only a single domain, which carries out the catalytic activity (and thus are monodomain, monomodular or even globular). Bearing a single domain is typical for lysins from Gram-negative-infecting phages [17]. However, many others, mainly from Gram-positives, are multidomain or multimodular. These latter kind of lysins may comprise domains devoted to catalysis and others whose function is substrate binding (a typical adaptation for enzymes that need to efficiently work on macromolecular, non-diffusible substrates [51]). Therefore, domains from lysins are either catalytic domains or cell wall-binding domains. The term “catalytic domain” seems to be traditionally the most used one, but with the expanding use in lysin engineering papers it became less functional. This is probably because of its ambiguous abbreviation, “CD”, which may be the reason why some groups are switching towards the term “enzymatically active domain”, with the more suitable abbreviation EAD. Similarly, cell wall-binding domains have been preferably abbreviated as “CBDs”. While this is an easy abbreviation, it can also cause confusion. For example, it can be mistaken for cellulose-binding domains or, more worrying, with choline-binding domains which are present, for example, in many pneumococcal phage lysins. Due to these ambiguities, the alternative abbreviation CWBD has also been used, especially in the context of pneumococcal lysins research, but it has not been widely adopted.

## 4. Names for Lysins as Antibacterial Agents

Once lysins were conceptualized as a new class of antibacterials in the context of the antibiotic resistance crisis, i.e., from the 2000s on, some new terms applied to lysins started to emerge. “Enzybiotics” is a name given to phage lysins purposed as antimicrobial agents. It is obviously a contraction of “enzyme-based antibiotics”, but the word misses the moiety “anti”, which has an essential meaning in “antibiotic” as “opposing life”. “Enzybiotic” is rather a term willing to indicate the use of enzymes as antibiotics by making up a similar word. This is to say, it would be a term applied to enzymes used to treat bacterial infections in a similar fashion as antibiotics. It was first coined by the V. Fischetti group [8], and since then it has been increasingly used, although “lysin” or “endolysin” seems to be much more popular still (Figure 1). While these latter terms are perfectly valid to refer to the enzymatic types of interest, the name “enzybiotic” indeed carries a more precise meaning as the molecule is explicitly applied as an antimicrobial. Therefore, not all lysins can be considered enzybiotics (or maybe they are all just potentially so) since not all lysins have been tested and purposed as antimicrobials. Also, the question of whether all enzybiotics can be considered lysins cannot be answered unequivocally. Since the main research on the repurposing of phage products as antimicrobials rely on lysins, the tendency is to implicitly equal “enzybiotic” to “lysin” [4]. Nonetheless, the term enzybiotic has also occasionally been applied to refer to the antimicrobial use of phage EPS depolymerases, and from a purely semantic perspective this seems correct to us given that depolymerases are enzymes and can be used in a similar fashion as antibiotics. Nevertheless, they act as antivirulence compounds rather than compounds that are directly bacteriostatic or bactericidal [52,53]. Some eccentric uses of this term are also documented, for example, the description of antimicrobial peptides (AMPs) as “membrane-targeted enzybiotics” [54], which would be, in our opinion, misleading, given the fact that AMPs are not, in principle, enzymes. Nevertheless, any other antimicrobial enzyme could be considered an enzybiotic, as has indeed already been pointed out [55]. This potentially includes phage lysins, bacterial lysins or autolysins (i.e., lysins produced by a bacterium to provoke its own lysis), phage or bacterial depolymerases, bacteriocins with an enzymatic action, or innate immunity lysozymes.

With the increasingly active research on lysin engineering to obtain better antimicrobial molecules, a plethora of names has been developed to distinguish between different engineering strategies. In fact, some efforts have already been made to provide a conceptual framework for the discussion of engineered lysins [56]. According to this framework, the first generation of lysins would be that of directly applied wild-type or natural lysins, while the second generation refers to lysins engineered to improve antibacterial and biochemical properties. There would be a third generation comprising those lysins engineered to enhance properties relevant under clinical conditions (bioavailability, half-life, interactions with the immune system, etc.). As a general term, synthetic lysin-based molecules are referred to as “engineered lysins”, regardless of whether they belong to the second or the third generation. Even more than mutagenesis, the development of novel engineered lysins relies on the principle of modular engineering, as explained in the previous section. This is how “chimeric lysins” (in a few cases also known as “chimeolysins”) started, first comprising only modules from peptidoglycan-degrading enzymes (both from phage and bacterial origin) and later incorporating functional modules from origins other than lysins too. An example of the latter are lysins fused to peptides able to permeabilize the outer membrane of Gram-negative bacterial cells and thus facilitate the antimicrobial activity of the peptidoglycan-degrading moiety. These are sometimes known as Artilysin^®^s, although this is a registered name by the company Lysando. Other engineering concepts focusing on outer membrane transfer have also been proposed, namely “innolysins” (a lysin fused to a phage receptor-binding protein or RBP [57] and lysocins (a lysin fused to a module from a bacteriocin) [58,59]. This latter name can be ambiguous since there is a new class of antibiotics that also go by the name of “lysocins” (Hamamoto, 2014).

## 5. Nomenclature of Individual Lysins

Another consequence of the newfound widespread interest in lysin research is the proliferation of different naming schemes used as common names for lysins. In the early stages of the molecular work on lysins, naming was rather functional and generally based on the original phage (e.g., T4 or T7 lysozyme, etc.). Further on, more “creative” names started to appear, mostly playing on the name and/or circumstances of the phage (e.g., Cpl-1 = “Complutense” phage lysin 1, being “complutense” the demonym of Alcalá de Henares, Spain, the place in which the original phage Cp-1 was isolated [60]). These non-standardized naming customs can still be found nowadays. But, more interestingly, efforts towards some standardization in naming schemes have also appeared. One of the most popular naming customs relies on the prefix “Lys”, referring to “lysin”, followed by the name of the phage, such as LysK (from *Staphylococcus* phage K [61]), LysGH15 (from *Staphylococcus* phage GH15 [62]), LysAB2 (from *Acinetobacter* phage phiAB2 [63]), and so on. An alternative is the prefix “Ply”, standing for “phage lysin”, normally also followed by the name of the phage itself (e.g., PlyC, PlySs2 [64,65]). The final traditional naming scheme for natural lysins would be that of using the name of the phage followed by the gene number, either preceded (OBPgp279, AP3gp15 [66,67]) or not (EL188, KZ144 [68]) by “gp”. These different traditions have become further complicated with the new engineering strategies. For engineered lysins there are, again, rather arbitrary names (such as Cpl-711, a chimeric lysin bearing the EAD of Cpl-7 and the linker and CBD from Cpl-1 [69]; or 1D10, which is the architectural variant at well D10 from screening plate 1 [70]) but also some standard ones. The prefix Cly is the counterpart of Ply for “chimeric lysins” (ClyS, ClyF, ClyJ [71,72,73]). Sometimes engineered lysins are the result of extracting domains from the original architecture, and this is noted in the names chosen, for example, for CHAP_K_ (*CHAP* domain from LysK [74]) or LysRODIΔAmi (LysRODI lacking its central amidase domain [75]). Naming based on the domain family name has also been used for chimeric lysins (e.g., CHAPSH3 [76]). Other names are dependent on the specific engineering strategy applied, such as the Art-### scheme for Artilysin^®^s [77]. In any case, it would be a daunting task to thoroughly review the varied naming rationales for lysins, and this paragraph is only meant to serve as an example. So, it will suffice to conclude that the diversity in naming schemes for engineered lysins undoubtedly makes it difficult to keep up with the research being produced these days in the field, and the adoption of either standard names or “barcodes” for newly derived engineered lysins might be desirable. For example, it is nowadays hard to retrieve precise information about engineered lysins: it takes a lot of time to find all necessary information in the Materials and Methods section of a paper about the exact nucleotide/amino acid sequences, while data management and sharing have become increasingly important. Such barcoding practices could help to readily identify the origin and engineering rationale without having to actually dive into the text of every new piece of literature. Complementarily, it would be good practice to submit all sequences of newly derived engineered lysins to publicly available databases (NCBI, EMBL, etc.) and make use of initiatives such as PhaLP, which aggregates phage lysin data from the main public databases and may be a starting point towards standardization [15]. This, in turn, has the potential to steer new discoveries based on automated analysis coupled to further standardized experimental data.

## 6. Concluding Remarks

Science aims at using well-defined, standardized names to conceptualize and refer to the objects it works with. As such, specific names are a tool essential to the communicative and generalizable aspect of science: accurate names allow effective discussion to take place at the global scientific agora. This is a good enough reason for taking some time and effort reflect on how nomenclature is used within our very own research field and maybe even on how it can become more uniform in the near future. Phage research already has a very long tradition. As it has become clear throughout this review, the names used in the field of phage lysins have changed linked to the respective historical circumstances. From our point of view, there have been at least three waves that have caused naming diversification and/or confusion with definitions around phage lysins:(1)The first one exemplified by the liberal use of the term “lysin” at the early stages, applied in general to a still undescribed agent causing lysis and thus confusingly referring to what later would be defined as either true endolysins, structural lysins, phage depolymerases, or even the phage itself. The driver for confusion at this stage was not precisely knowing the nature of what was actually causing cell lysis (and, to a certain point, not even grasping the nature of the phage);(2)A second one driven by the initial lack of evidence or tools able to precisely define biochemical specificity, which at some point derived from the naming of lysins in general as “lysozymes”, a tradition that survived even after the name “lysozyme” was reserved for a precisely described catalytic mechanism of peptidoglycan hydrolysis;(3)A third one is our current era of engineered lysins, which is being driven by a variety of techniques or conceptualizations, and whose result is a remarkable diversification both in particular names for engineering strategies and naming schemes for individual lysins.

To these three, a fourth one may be added on top if we consider the names developed by companies devoted to phage lysin development for the purpose of market positioning. These range from general names for their lysin-based products (DLAs = “direct lytic agents”, EPLEs = “engineered phage lytic enzymes”) to new naming schemes for individual molecules (e.g., “Exebacase”, relying on the enzymatic suffix “-ase”, as recommended by the WHO for the International Nonproprietary Names [INNs] for enzymatic pharmaceuticals [78]).

At this point, we believe efforts toward correct name use and standardization should be welcomed to promote sharing and profit from the huge amount of data being produced around the world. For further clarity, a provisional glossary of phage lysin-related names discussed in this review can be found in Appendix B. As for naming standardization, some global initiatives already in place may be relevant to the topic, such as the enzymatic names and definitions by the International Union of Biochemistry and Molecular Biology (IUBMB, accessible here: https://www.enzyme-database.org (accessed on 5 August 2023)) or the ICTV position on the naming of phages [79], which could be used as an example to develop structured names for engineered lysins. To help initiate a fruitful discussion, a few key intervention points extracted from this article can be found in Appendix C. This may be a starting point for what we believe is an obligation of the whole lysin community towards achieving a widely supported nomenclature.

## Figures and Tables

**Figure 1 cells-12-02016-f001:**
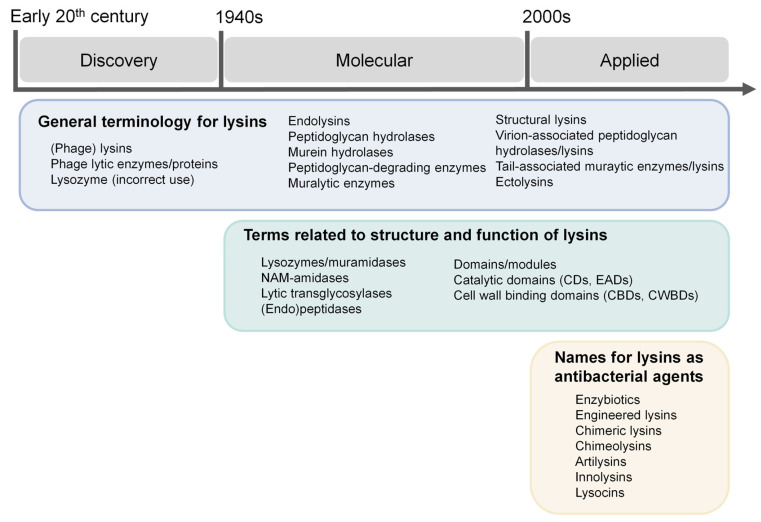
Conceptual outline for the discussion of nomenclature in the phage lysins field.

**Figure 2 cells-12-02016-f002:**
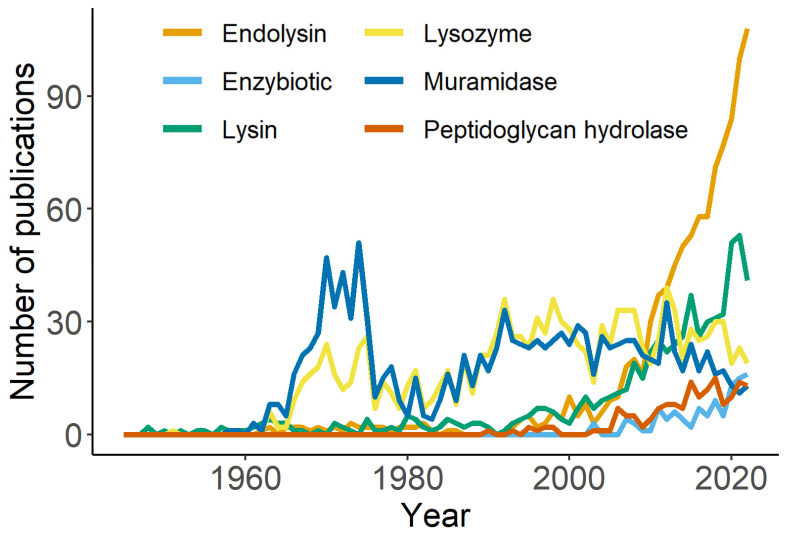
Total references per year according to search terms as described in Table 1.

**Table 1 cells-12-02016-t001:** Number of references found by PubMed searches regarding relevant phage lysin-related terms. ^a^ As found in the indicated PubMed searches (performed on 26 May 2023) including, when available, article titles, abstract, whole text and/or MeSH terms.

Name	Number of References ^a^	Search
Lysin	608	((“lysin” OR “lysins”) AND (phage OR bacteriophage))
Endolysin	934	((“endolysin*”) AND (phage OR bacteriophage))
Peptidoglycan-degrading enzyme	15	((“peptidoglycan-degrading enzyme*”) AND (phage OR bacteriophage))
Muralytic enzyme	26	((“muralytic enzyme*”) AND (phage OR bacteriophage))
Cell wall degrading enzyme	25	((“cell wall degrading enzyme*”) AND (phage OR bacteriophage))
Muralysin	3	“Muralysin”
Peptidoglycan hydrolase (PGH)	150	((“peptidoglycan hydrolase*”) AND (phage OR bacteriophage))
Murein hydrolase	53	((“murein hydrolase*”) AND (phage OR bacteriophage))
Phage lytic protein	20	“phage lytic protein*”
Phage depolymerase	193	((depolymerase) AND (phage OR bacteriophage))
Phage exopolysaccharide depolymerase	18	((“exopolysaccharide depolymerase*” OR “EPS depolymerase*”) AND (phage OR bacteriophage))
Lysozyme	1202	“lysozyme*” AND (phage OR bacteriophage)
Muramidase	1215	“muramidase*” AND (phage OR bacteriophage)
N-acetylmuramoyl (NAM-) amidase	388	(“amidase*” OR “NAM-amidase*”) AND (phage OR bacteriophage)
Endopeptidase	500	((“endopeptidase*”) AND (phage OR bacteriophage) AND (lysin OR endolysin))
Peptidase	80	((“peptidase*”) AND (phage OR bacteriophage) AND (lysin OR endolysin))
Glucosaminidase	21	((glucosaminidase) AND (phage OR bacteriophage))
Lytic transglycosylase	53	((lytic transglycosylase) AND (phage OR bacteriophage))
Virion-associated peptidoglycan hydrolase (VAPGH, VAPH)	21	“virion-associated peptidoglycan hydrolase*”
Virion-associated lysin	3	“virion-associated lysin*”
Enzybiotic	99	“enzybiotic*”
Engineered lysin	11	“engineered lysin*”
Engineered endolysin	15	“engineered endolysin*”
Chimeric lysin	35	“chimeric lysin*”
Chimeric endolysin	12	“chimeric endolysin*”
Chimeolysin	8	“chimeolysin*”
Innolysin	2	“innolysin*”
Lysocin	1	lysocin* AND (phage or bacteriophage) NOT “lysocin E”
Artilysin	12	“artilysin*”
DLA	10	“direct lytic agent*”
EPLE	1	“engineered phage lytic enzyme*”
Module	78	“module*” AND (lysin OR endolysin) AND (phage or bacteriophage)
Domain	452	“domain*” AND (lysin OR endolysin) AND (phage or bacteriophage)
Enzymatically active domain (EAD)	23	“enzymatically active domain*” AND (lysin OR endolysin) AND (phage or bacteriophage)
Catalytic domain (CD)	140	“catalytic domain*” AND (lysin OR endolysin) AND (phage or bacteriophage)
Cell wall-binding domain (CBD, CWBD)	137	“cell wall binding domain*” AND (lysin OR endolysin) AND (phage or bacteriophage)

## Data Availability

No new data was created or analyzed in this study. Data sharing is not applicable to this article.

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
