# Peer review of "What’s in a Name? An Overview of the Proliferating Nomenclature in the Field of Phage Lysins"

_cells, 2023, doi:10.3390/cells12152016_

Round 1

Reviewer 1 Report

The review by Vázquez and Briers provides both a history on phage lysin research and a discussion point for those currently working in the field of phage lysins. In a similar way that bacteriophage taxonomy has recently undergone an overhaul, partly due to ever increasing availability of sequence data, the authors here also propose the standardisation of the nomenclature around phage lysins, which will hopefully benefit all those with an interest in lysins. Overall, the manuscript is very well written and researched.

The authors do a commendable job of describing approximately 100 years of lysin related research. The review is well structured and easy to digest by virtue of organising this huge body of work into three distinct ‘eras’ of phage research and by applying a thematic review to the terminology used. This includes a comprehensive review of the literature and the chronology of terms associated with lysins. The etymology of these terms is well explained and their rises and falls in relative popularity are contextualised against the relevant phage research which was taking place at the time.  

The potential need for standardising the naming of phage lysins is clearly justified. The review provides excellent examples of this e.g. misnaming, which can infer incorrect catalytic properties (as is the case with some ‘lysozymes’ and the use of other generic terms like peptidoglycan hydrolases which stricto sensu should not include lyases). Then there is the dizzying array of new terms used when describing lysins as therapeutics. As such, this review provides a timely discussion point for researchers to consider implementing agreed nomenclature.

Navigating these issues will not be straightforward. In some instances, the authors offer suggestions for what they believe should be accepted principles e.g. names based on catalytic activity should only be used where empirical evidence exists, correct use of the term ‘domains’ and ‘modules’ etc. These suggestions seem sensible and could begin to provide a framework of standardisation for the field. Therefore, my main suggestion is that the authors perhaps provide a summary of their suggestions for standardisation of nomenclature in the concluding remarks.  This seemed to be one of the main aims of the review. I understand that the authors would like the community to come together and agree on these, however providing even a basic blueprint may help begin the discussion?

My other comment is that sub-headings 4 and 5 are the same ‘Names for lysins as antibacterial agents’ – is it necessary to have both? Perhaps the first could be ‘generic names for lysins as bacterial agents’ and the second as ‘nomenclature of individual lysins used as antibacterial agents?’

Author Response

We thank the reviewer for their assessment and we have implemented their suggestions, which we indeed think strengthen the purpose of the manuscript:

  • Sections 4 and 5 are now correctly named.
  • And additional paragraph has been added to the concluding remarks to summarize the main suggestions that can be derived from our work to serve as a starting point for the discussion.

Reviewer 2 Report

The proposed review by Roberto Vázquez and Yves Briers is of great interest, given the significant number of ambiguities in the way phage endolysins are referred to. This review is, therefore, more than welcome. Moreover, it is well-written and methodically covers the existing literature in the field. I highly recommend its publication in its present form after the following modification: The chapter heading "Names for lysins as antibacterial agents" at line 308 is the same as that at line 361.

Author Response

We thank the reviewer for their kind assessment. In the revised version, we have fixed the repeated titles of sections 4 and 5.